# Size Matters? A Comprehensive In Vitro Study of the Impact of Particle Size on the Toxicity of ZnO

**DOI:** 10.3390/nano13111800

**Published:** 2023-06-04

**Authors:** Montserrat Mitjans, Laura Marics, Marc Bilbao, Adriana S. Maddaleno, Juan José Piñero, M. Pilar Vinardell

**Affiliations:** 1Physiology, Department of Biochemistry and Physiology, Universitat de Barcelona, 08028 Barcelona, Spain; montsemitjans@ub.edu (M.M.); laura.marfa.89@gmail.com (L.M.); m.bilbasen@gmail.com (M.B.); adrianamaddaleno@ub.edu (A.S.M.); juanjopinero@ub.edu (J.J.P.); 2Institute of Nanoscience and Nanotechnology, Universitat de Barcelona, 08028 Barcelona, Spain

**Keywords:** ZnO, nanoparticle size, in vitro, cytotoxicity, hemolysis, protein corona

## Abstract

This study describes a comparative in vitro study of the toxicity behavior of zinc oxide (ZnO) nanoparticles and micro-sized particles. The study aimed to understand the impact of particle size on ZnO toxicity by characterizing the particles in different media, including cell culture media, human plasma, and protein solutions (bovine serum albumin and fibrinogen). The particles and their interactions with proteins were characterized in the study using a variety of methods, including atomic force microscopy (AFM), transmission electron microscopy (TEM), and dynamic light scattering (DLS). Hemolytic activity, coagulation time, and cell viability assays were used to assess ZnO toxicity. The results highlight the complex interactions between ZnO NPs and biological systems, including their aggregation behavior, hemolytic activity, protein corona formation, coagulation effects, and cytotoxicity. Additionally, the study indicates that ZnO nanoparticles are not more toxic than micro-sized particles, and the 50 nm particle results were, in general, the least toxic. Furthermore, the study found that, at low concentrations, no acute toxicity was observed. Overall, this study provides important insights into the toxicity behavior of ZnO particles and highlights that no direct relationship between nanometer size and toxicity can be directly attributed.

## 1. Introduction

Nanomaterials (NMs) are one of the most versatile products, but because of their intrinsic physicochemical properties, they can interact with various biological molecules or cells, causing alterations in the organism. ZnO nanoparticles exhibit promising biomedical applications based on their biological activities [1]. Nevertheless, when in contact with living organisms, nanoparticles interact with proteins, bind to them to form a protein corona, and modulate their biological activity [2]. In vitro studies offer the possibility of evaluating such NMs before conducting more complex and toxicological preclinical studies. In vitro methods have the capacity to evaluate the potential hazards related to the administration of nanoparticles and understand the mechanisms of adverse effects.

ZnO nanoparticles (NPs) have been extended to be used as biosensors [3]; cosmetics [4,5]; optical devices [6]; and different biomedical applications [7], such as drug delivery [8,9], among others. However, there are still concerns about their potential risk to human health. Thus, the study of their cytotoxic effects is of interest.

The interactions of nanomaterials with blood or blood components are preliminary steps in characterizing their cytotoxic profiles. However, the composition of the media, the temperature, and other fundamental parameters can lead to erroneous conclusions [10]. Here, we study the influence of different parameters on the potential hemolysis induced by commercial ZnO micro- and nanoparticles (<50 nm and <100 nm) and the effect of these materials on the coagulation process.

It is known that colloid suspensions tend to adsorb proteins on their surface, leading to the formation of a superficial layer known as the protein corona. The protein corona can alter the physiological properties of NPs and their interactions with other molecules once in contact with the biological environment [11]. Depending on the NP size and chemistry, the protein can have a different affinity for the NP surface. The proteins adsorbed on NPs can affect the aggregation, transference, and intracellular uptake of NPs [12]. As serum albumin and fibrinogen are the two most abundant proteins in the corona formed on both bulk- and nano-ZnO [13], it is necessary to study the influence of these two proteins. In this sense, we have studied the influence of different particle sizes (micrometric, <50 nm, and <100 nm) of commercially available ZnO on the formation of protein coronas and coagulation assays.

To study the toxicity of nanoparticles in vitro, it is important to select an appropriate cell type considering the route of entry into the body and the most relevant cytotoxicity assay [14,15]. In this sense, the purpose of this work is also to evaluate the potential cytotoxic effects of ZnO micro- and nanoparticles on two human cell lines. The cell lines studied are HaCaT as a representative of normal cells of the epidermis, considering topical administration, and A549 as a model of lung epithelium for potential aerial penetration. The cytotoxicity was evaluated using three different endpoints, namely, MTT, NRU, and LDH.

This comparative in vitro cytotoxic study will increase knowledge of how metal oxide nanoparticles develop adverse effects.

## 2. Materials and Methods

### 2.1. ZnO Particles and Reagents

ZnO particles in nano- (50 nm and 100 nm) and micro-forms were purchased from Sigma-Aldrich (St. Louis, MO, USA). Products are defined by the manufacturer as ZnO nanopowders with a <50 nm particle size and a <100 nm particle size, both with a purity of >95%, and micro-ZnO is defined as an ACS reagent of 99%.

Al_2_O_3_ powder (99.8%) and nanopowder (13 nm, 99.8%) were also obtained from Sigma-Aldrich (St. Louis, MO, USA).

Phosphate-buffered saline at pH 7.4 was prepared in house with extra-pure potassium dihydrogen phosphate, synthesis-grade sodium chloride (Scharlau, Sentmenat, Spain), and di-sodium hydrogen phosphate anhydrous (Panreac, Castellar del Vallés, Spain).

For SDS-PAGE electrophoresis, acrylamide (40%), bisacrylamide (2%), tetramethylethylenediamine (TEMED), ammonium persulfate, β-mercaptoethanol, and bromophenol blue were obtained from GE Healthcare Bio-Sciences AB (Uppsala, Sweden); sodium dodecyl sulfate (SDS) was purchased from (Sigma-Aldrich, Sant Louis, MO, USA); and Precision Plus Protein™ Standard (ranging from 10 to 250 kDa) was obtained from Bio-Rad Laboratories, Inc. (Hercules, CA, USA).

Dimethylsulfoxide (DMSO), neutral red solution, thiazolyl blue tetrazolium bromide (MTT), and trypan blue solution 0.4% were purchased from Sigma-Aldrich (St. Louis, MO, USA). Catalyst solution (Diaphorase/NAD+) and dye (INT/Na-lactate) were obtained from Takara, Clontech Laboratories (Kyoto, Japan).

Bovine serum albumin (≥96%) and fibrinogen (50–70% protein; ≥80% of protein is clottable) were acquired from Sigma-Aldrich (Sant Louis, MO, USA, ≥96%); Coomassie brilliant blue G-250 (BioRad); D(+)-sucrose was obtained from Carlo Erba Reagents (RPE-for analysis-ACS).

### 2.2. Characterization of ZnO Particles

#### 2.2.1. Atomic Force Microscopy (AFM)

To characterize the topography and particle size of the ZnO nanoparticles, a droplet (about 40 μL) of a ZnO-PBS solution (100 μg/mL) was deposited on a previously cracked mica substrate. Then, samples were quickly dried with N_2_ and finally observed with a Bruker Nanoscope V Multimode 8 (USA) at the Centres Científics i Tecnològics of Universitat de Barcelona (CCiT UB).

#### 2.2.2. Transmission Electron Microscopy (TEM)

In this study, ZnO nanoparticles of 50 nm or 100 nm were suspended at a final concentration of 100 μg/mL in PBS, 0.9% NaCl, and ethanol. The suspension was obtained from the dilution and ultrasonication of a previously ultrasonicated suspension of 1.0 mg/mL of the sample. Observation and analysis were carried out using a JEOL JEM LaB6-2100 microscope with one drop (5 µL)) deposited and dried on a Holey Carbon-Cu grid (CCiT UB).

#### 2.2.3. Dynamic Light Scattering

The mean hydrodynamic diameter (HD) and the polydispersity index (PDI) of the particles were determined with dynamic light scattering (DLS) using a Malvern Zetasizer ZS (Malvern Instruments, Malvern, UK) at a scattering angle of 173° and with a refractive index of 2.003 specific to ZnO. Briefly, the metal oxide particles, at a final concentration of 1.0 mg/mL, were incubated in phosphate-buffered saline (PBS, pH 7.4), PBS containing albumin or fibrinogen (2 mg/mL), or DMEM 5% FBS for 2 h and 24 h at 37 °C. Each measurement, made with a 10 mm × 10 mm quartz cuvette (Hellma Analytics, Müllheim, Germany), consisted of 3 series of 5 readings. Finally, results were expressed as the mean of three independent experiments.

In addition, after incubation, samples were centrifuged (15,300× *g* 5 min), and the amount of protein in the supernatant was determined with the Bradford assay [16] to calculate the protein adsorbed by subtracting from the amount determined before incubation using bovine serum albumin (BSA) as the standard.

### 2.3. Study of ZnO Albumin Interactions by Spectrometric Analysis

The interaction between ZnO particles of different sizes and albumin was investigated using bovine serum albumin (BSA) at physiological pH (7.4). A stock solution of BSA was prepared at a concentration of 4 mg/mL; different concentrations of ZnO were added to 3 mL of BSA to obtain a final concentration of 2 mg BSA/mL. This mixture was homogenized and kept for 30 min for incubation. The same assay was repeated with a previous incubation of 18 h. After incubation, UV-visible absorption spectra were recorded in a wavelength range of 250–330 nm (Shimadzu UV-Vis 160, Kyoto, Japan) using a one-centimeter-path-length quartz cuvette [17].

Moreover, fluorescence measurements (Luminescence Spectrometer, Aminco Bowman) were recorded in a wavelength range of 310–430 nm using a one-centimeter-path-length quartz cuvette. The excitation wavelength was set at 278 nm and emission was measured in a range of 275–475 nm [17].

### 2.4. Hemocompatibility Studies

#### 2.4.1. Blood Collection and Preparation of Erythrocyte Suspension

Fresh human blood was extracted from healthy volunteers via venopunction and drawn into tubes containing EDTA or sodium citrate after informed consent and approval of the Bioethics Committee of the University of Barcelona, Spain (protocol code IRB00003099, 15 February 2021).

Blood extracted with EDTA was centrifuged for 10× *g* min at 3000 rpm to remove plasma traces, buffy coat, and platelets, leaving only erythrocytes. Immediately, the isolated red blood cells (RBC) or erythrocytes were suspended in twice the volume of saline solution (PBS pH = 7.4), and the washing procedure (centrifugation/discarding plasma/washing with PBS) was carried out three more times until a transparent and colorless supernatant was obtained. The final RBC suspension was adjusted by adding adequate PBS to obtain a maximal hemoglobin absorbance at 575 nm of 1.8–2.1.

#### 2.4.2. Potential Media Interaction on Hemoglobin Adsorption

Before studying the hemolytic behavior of ZnO particles, the potential existence of interactions with hemoglobin and if these interactions were due to pH, the media composition, and the metal oxide nanoparticles studied were investigated [18]. Based on the isoelectric point of hemoglobin protein (6.8), studies were performed at pH 7.4 and 5.7 in PBS, 0.9% NaCl 0.9% solution, and tris(hydroxymethyl)aminomethane–maleate (tris–maleate) buffer solution. Furthermore, Al_2_O_3_ nanosized particles were chosen for comparative purposes.

PBS solution (pH 7.4) was acidified with HCl 0.1 M to obtain PBS at pH 5.7, whereas, in the case of 0.9% NaCl (pH 5.7), the addition of NaOH 0.1 M increases the pH to 7.4. Finally, a tri–maleate buffer solution was obtained by mixing a 0.2 M solution of tris acid maleate aminomethane (A) with 0.2 M of NaOH (B). The appropriate amount of each solution to obtain the desired pH values is provided by the following formula:50 mL A + X mL B diluted at a final volume of 200 mLpH 5.7, B = 15.5 mLpH 7.4, B = 54.0 mL(1)

Adsorption of hemoglobin was assessed by incubating a fixed concentration of the metal oxides (1.0 mg/mL) with an aliquot of human erythrocyte suspension or human hemoglobin solution for 30 min at room temperature (RT) in the presence of different media. Then, samples were centrifuged at 10,000× *g* rpm for 5 min (Nahita Blue, high-speed centrifuge 2624/2, Sudelab S.L., Spain), and finally, the supernatants of the samples were qualitatively observed to assess if the hemoglobin was adsorbed by the metal oxide.

#### 2.4.3. Hemolytic Activity: Effect of Temperature and Albumin

To study the membrane-lytic activity, various concentrations of ZnO oxide particles (0.2 mg/mL–1.6 mg/mL) at a final volume of 1 mL were prepared from freshly prepared stock solutions of 10 mg/mL in PBS. Then, an aliquot of 25 μL of the erythrocyte suspension was introduced to each tube, and the tubes were placed in a rotary shaker for incubation at RT or 37 °C for a period of 24 h. At the end of incubation, samples were centrifuged (10,000× *g* rpm for 5 min) in a Nahita Blue high-speed centrifuge 2624/2 (Sudelab S.L., Spain), and supernatant absorption was determined at 540 nm (Shimadzu UV-Vis 160, Japan). Each experiment was repeated around three times in triplicate, and appropriate controls consisting of RBC incubated in PBS pH 7.4 (negative control or basal hemolysis) and deionized water (positive control or total hemolysis) were included [19].

The percentage of RBC hemolysis was calculated using the following formula [20]:Hemolysis % = [*A_S_* − *A_C_*]/[*A*_100_ − *A_C_*] × 100,(2)
where *A_S_* is the absorbance obtained for each sample, *A_C_* is the absorbance of the negative control (basal hemolysis), and *A*_100_ is the absorbance of the positive control or total hemolysis (100%).

The potential protective effect of albumin on hemolysis was also studied. It is well established that proteins from body fluids bind to NP surfaces upon their exposure to an organism, forming a so-called “protein corona” around the NPs [21]. Albumin is the most abundant protein in the human body. Moreover, previous studies have demonstrated that the addition of 0.5 mg/mL of albumin to PBS (pH 7.4) decreases the hemolytic behavior of Al_2_O_3_ nanoparticles [22]. In this case, negative controls (*A_C_*) consisted of RBC incubated in pH 7.4 PBS containing 0.5 mg/mL of albumin.

#### 2.4.4. Effects of ZnO on Erythrocyte Morphology

An evaluation of ZnO particles over erythrocyte morphology was conducted using Scanning electron microscopy (SEM). Similarly, as in Section 2.4.3. RBCs were exposed to 0.2 mg/mL of each ZnO particle or PBS as a morphology control and incubated for 3 or 24 h at RT. Then, samples were centrifuged at high speed (10,000× *g* rpm for 5 min), the supernatant was discarded, and 1 mL of 2.5% glutaraldehyde in PBS was added. Fixed samples were washed with PBS solution, post-fixed with 1% osmium tetroxide, placed over a glass coverslip, dehydrated in an ascending series of ethyl alcohols (50–100%), air-dried using the critical point drying method in a CPD 7501 apparatus (Polaron, Watford, UK), and finally mounted on a carbon stub. The resulting specimens were observed at CCiT UB using a Hitachi 2300 electron microscope operating at 15 kV.

#### 2.4.5. Coagulation Assays

Fresh human plasma was obtained from whole blood extracted with sodium citrate (see Section 2.4.1. and posteriorly centrifuged to separate plasma from cellular elements. Then, plasma was incubated with each of the three ZnO particles for a period of 30 min at 37 °C under soft rotation before evaluating the effects on coagulation time [23]. Effects on coagulation were studied by determining the prothrombin time (PT) using RecombiPlasTin 2G (HemosIL kit, Werfen, Spain) and the activated partial thromboplastin time (aPTT) using SynthASil (HemosIL kit, Werfen, Spain) following the protocol provided by the manufacturer.

Different concentrations of ZnO particles were assessed in each run, namely, 0.1, 0.5, and 1.0 mg/mL in quadruplicate. In each independent experiment, control samples consisting of plasma incubated with a vehicle (PBS, pH 7.4) were included. Each experiment was repeated around three times with plasma from different donors.

Finally, protein adsorption onto ZnO particles after incubation with human plasma was quantitatively and qualitatively evaluated in the pellets using the Bradford assay, SDS-PAGE, and TEM (transmission electron microscopy).

After incubating the ZnO particles in human plasma, the adsorption of proteins was also explored with TEM. For this reason, before observation, samples were diluted to a final concentration of 0.02 µg/mL in H_2_O (MiliQ quality) and dyed with phosphotungstic acid to reveal the presence of protein. Observation and analysis were carried out using a JEOL JEM LaB6-2100 microscope with one drop deposited and dried on a Holey Carbon-Cu grid. A complementary analysis of elemental composition using energy-dispersive X-ray spectroscopy (EDX) was performed to confirm the presence of plasma proteins, ZnO particles, and the staining agent.

#### 2.4.6. Two-Dimensional Gel Electrophoresis of Human Plasma Protein: SDS-PAGE

As a preliminary study, the different proteins involved in the protein corona SDS-PAGE experiments were examined. Thus, human plasma samples were incubated for 30 min at 37 °C with ZnO particles at a concentration 10 times higher than those used in Section 2.4.3. and then centrifuged (13,000× *g* rpm, 5 min.) to separate the pellet from the supernatant and study each of them separately. Pellets were washed three times with a solution of sucrose 0.7 M in PBS to facilitate pellet separation from the supernatant [24]. Finally, the amount of protein in both supernatants and pellets was determined using the Bradford assay, as previously stated [25]. Controls without particles were included for qualitative comparative purposes.

The following process was used to perform SDS–polyacrylamide gel electrophoresis on 18 micrograms of extracted protein: the method reported by [26] was followed, and a Mini-PROTEAN Tetra Cell unit from Bio-Rad (Hercules, CA, USA) was used. The gel used consisted of two parts: a 7.5% polyacrylamide resolving gel and a 5% polyacrylamide stacking gel. The extracted protein was mixed with 2× SDS sample buffer, heated for 5 min at 95 °C, and then loaded onto the gel. Electrophoresis was run for 10 min at 60 V and then continued for 40 min at 200 V. The protein bands were stained with Coomassie Blue R-250 for 40 min while gently shaken, and then, the excess stain was removed using a mixture of 7.5% methanol and 10% acetic acid. The molecular weight of plasma proteins was estimated based on a molecular size marker ranging from 10 to 250 kDa.

### 2.5. Cell Culture and Cytotoxicity Studies

Two cell lines were studied: a human keratinocyte cell line, HaCaT (Eucellbank, Barcelona), and the A549 line, originating from the lung epithelium (European Collection of Cell Cultures, EACC, Salisbury, England). Cells were grown and maintained in DMEM medium (4.5 g/L glucose) supplemented with 10% (*v*/*v*) FBS (Fetal Bovine Serum), 2 mM L-glutamine, 100 U/mL penicillin, and 100 µg/mL streptomycin in an incubator at 37 °C, 5% CO_2_. All culture reagents were purchased from Lonza (Verviers, Belgium) except FBS HyClone^®^, which was acquired from Thermo Scientific (Northumberland, UK).

Briefly, when cultures reached 80% confluence, the culture medium was discarded and cells were detached with trypsin-EDTA (Lonza, Verviers, Belgium). From the cell suspension obtained, cell density was determined in an aliquot treated with the vital dye trypan blue (Sigma-Aldrich, Madrid, Spain). Cell density was adjusted to 1 × 10^5^ cells/mL, and cells were seeded in 96-well plates and maintained for 24 h in an incubator at 37 °C, 5% CO_2_. Then, cytotoxic activity was investigated by exposing cells for 24 h to different concentrations of ZnO particles ranging from 0.78 to 100 µg/mL prepared from a freshly prepared solution of 1.0 mg/mL [27] in DMEM medium with 5% FBS, 2 mM L-glutamine, and 1% antibiotic. Before applying the treatment, the medium was aspirated, and then, the cells were exposed to the compounds in decreasing concentrations. For each plate and experiment, untreated cells remaining in the DMEM medium as controls of viability were included. Finally, the plates were incubated for 24 h at 37 °C and 5% CO_2_. The assessment of the cytotoxic potential of ZnO particles was performed using three different methods: lactate dehydrogenase (LDH) determination, reduction of 2,5 diphenyl-3-(4,5-dimethyl-2-thiazolyl) tetrazolium bromide (MTT) dye, and the neutral red uptake (NRU) assay.

To avoid surface nanoparticle adsorption, LDH was detected in cell-free supernatant without previous centrifugation [28]. Thus, the supernatant of each sample was introduced o a clear plate and mixed with the same volume of the reaction mixture following the instructions of a commercially available kit (Takara, Japan). The plate was incubated at room temperature and protected from light for up to 30 min, and then, absorbance at 492 nm was measured with a Tecan Sunrise microplate reader (Männedorf, Switzerland). Viable cells were calculated as a percentage of cell viability according to the following formula:Cell viability % = 100 − [(([*A_C_* − *A_B_*]/[*A_T_* − *A_B_*]) − 1) × 100](3)
where *A_C_* is the absorbance of untreated cells, *A_B_* is the background absorbance (medium without cells), and *A_T_* is the absorbance obtained for treated cells.

In the case of MTT, we followed the protocol of Mosmann [29] with slight modifications [30]. Thus, after extracting the supernatant, 100 µL of a solution of 0.5 mg/mL MTT in serum-free DMEM without phenol red was introduced to each well, and the plate was incubated for at least 3 h at 37 °C and 5% CO_2_. The formazan crystals formed were dissolved in 100 µL of dimethyl sulfoxide (DMSO), and the plate was agitated for 10 min at 100 rpm/min to homogenize the well content. For the uptake of NR, steps to the MTT were followed according to Babich and Borenfreund [31]. In this case, a 0.05 mg/mL NR in serum-free DMEM without phenol red solution was used, and a destain solution containing an acidic ethanol solution was added to dissolve the NR uptake by viable cells. Finally, absorbance was measured at 550 nm with a Tecan Sunrise microplate reader (Männedorf, Switzerland), and cell viability was calculated as a percentage considering that untreated cells had 100% viability, as indicated by the following formula:Cell viability % = [*A_T_* − *A_B_*]/[*A_C_* − *A_B_*] × 100(4)
where *A_C_* is the absorbance of untreated cells, *A_B_* is the background absorbance (medium without cells), and *A_T_* is the absorbance obtained for treated cells. Potential interferences of ZnO particles in the assay reagents and dyes were previously discarded by incubating them in the same conditions but without cells.

### 2.6. Statistical Analysis

Each experiment was repeated at least three times, including triplicates for each condition, and results are expressed as the mean ± standard error. An unpaired Student’s *t*-test or a one-way analysis of variance (ANOVA) were used to determine the differences between the data sets, followed by a Scheffé post hoc test for multiple comparisons using the SPSS^®^ program (SPSS Inc., Chicago, IL, USA) Differences were considered significant at *p* < 0.05, as indicated in the figure and table footnotes.

## 3. Results and Discussion

### 3.1. Characterization of ZnO Particles

#### 3.1.1. Atomic Force Microscopy (AFM)

Topographical images of the ZnO nanoparticles are shown in Figure 1. From the AFM characterization, it was possible to determine the relative abundance of ZnO particles of different sizes and the mean size: 373.5 ± 114.2 nm for the ZnO 50 nm nanoparticles and 173.2 ± 43.8 for the 100 nm nanoparticles. Thus, the particle sizes of ZnO provided by AFM differ from those provided by the manufacturer, namely, those of 50 nm. These much higher values indicate the tendency of nanoparticles to aggregate when they are put in a solution. In accordance with their smaller size, this tendency is much more important for 50 nm particles.

#### 3.1.2. Transmission Electron Microscopy (TEM)

TEM is a very powerful technique to study nanosized systems since it allows for the analysis of size and crystallographic characteristics [32]. However, here, we used TEM images to qualitatively analyze the behavior of ZnO nanoparticles in different media and compare them with AFM outcomes. In this sense, Figure 2 shows that, in the case of PBS (A and D), NPs formed great polyhedral aggregates as indicated previously by AFM images, confirmed by their chemical nature with energy-dispersive X-Ray spectroscopy and selected area diffraction analysis (Appendix A). Moreover, these observations agree with previous studies reporting that, in physiological media, NPs have a tendency to agglomerate to form micro-sized particles that are more stable in such environments [33].

When the nanoparticles were suspended in 0.9% NaCl (B, E), it was observed that newly formed aggregates at the microscale were also formed. These aggregates are bigger for ZnO 50 nm than 100 nm ZnO particles and qualitatively smaller than those formed in PBS (about 3 to 5 times). Finally, the NPs were observed dissolved in ethanol to determine the real size of the particles. It was thought that, because this medium is very volatile, no aggregation would occur therein. However, as shown in images C and F of Figure 2, aggregates of ZnO NPs were also observed, although a smaller size than those observed in the other two media. Moreover, similarly to what happens in the saline 0.9% NaCl medium, the 50 nm ZnO forms larger aggregates than the 100 nm ZnO, as was also observed here with the AFM characterization.

#### 3.1.3. Dynamic Light Scattering (DLS)

The population of ZnO NPs in PBS solution, as measured with the dynamic light scattering method, was heterogeneous in size because of the formation of large aggregates [34]. Consequently, this method measures the sizes of the clustered particles rather than individual particles [32].

The hydrodynamic diameter (HD) obtained using DLS on ZnO particles in PBS is presented in Table 1. As observed, the HD obtained at twenty-four hours is greater than at two hours, indicating that, to a greater or lesser degree, particles agglomerate [2,27], and such agglomerates at two hours are fifteen to twenty times larger than the primary particles [33]. When considering HD values after 10 min of incubation (Appendix A), as well as AFM and TEM observations, it can be concluded that NPs aggregate immediately after being introduced to physiological media.

We also observed that ZnO 50 nm tends to form larger aggregates than 100 nm ZnO and that using other media such as 0.9% NaCl or ethanol does not prevent the tendency of NPs to agglomerate. SEM images of 100 nm ZnO obtained by Jung et al. in 2021 also demonstrated this aggregation but allowed for the determination of a particle size that resembled the one described by the provider [35].

These measurements have the added problem of the fast sedimentation of the NPs, but they agree with those obtained with the previous techniques, showing the strong agglomeration of the ZnO particles [34]. For this reason, the hydrodynamic size of ZnO particles is only qualitatively relevant because of the extremely high polydispersity index that we obtained (from 0.426 to 0.851). It can be concluded that, when immersed in aqueous media, homogeneous suspensions are unstable, and our particles quickly agglomerated and form randomly sized clusters. All results obtained to characterize the NPs in PBS indicate that the behavior of metal oxides in biological media is poorly predictable since they aggregate, agglomerate, and even degrade in neutral media.

#### 3.1.4. Influence of Proteins on Hydrodynamic Diameter

Jung et al. [35] indicate that the presence of proteins increases the hydrodynamic size of NPs; then, we compared values obtained with different techniques: SEM and DLS. Therefore, the effect of proteins on nanoparticle size is investigated using DLS, and, specifically, we studied BSA and fibrinogen, as well as DMEM 5% FBS (Table 2).

In the case of proteins, our results reveal that, similarly to those described for PBS, the HD increases with time when BSA is present in the medium. However, HD values are, in general, statistically significantly lower than those shown in Table 1 and, in the case of fibrinogen, attain the lowest values recorded. This decrease in size can be attributed to the effect of preventing the aggregation of particles, providing greater stability to the system [36]. However, at 24 h, this stabilization is basically attributed to fibrinogen, as the case of BSA and, is only observed for ZnO at 50 nm. In the case of DMEM, there is also an increase in the HD with time, except for micro-sized particles, and values are statistically significantly lower than in the case of PBS but higher than those reported previously by other authors [27,33,35].

In summary, we can conclude that similar behavior was observed between the three different particle sizes: the maximum particle size and increase in HD are determined in PBS, then in BSA and DMEM, and finally in fibrinogen. Regarding micro-sized ZnO, there are certain discrepancies with its behavior because its HD seems most affected by the presence of BSA. The reason remains that micrometric materials compared with nanometric ones have different properties [33].

The quantification of total protein after incubation with BSA and fibrinogen was performed using the Bradford method, and finally, the percentage of protein adsorbed was calculated. Figure 3 reveals that the percentage of protein adsorption after incubation is very low in the case of BSA (up to 2.7%), while this percentage is considerably higher for fibrinogen at 24 h [2]. In this latter case, the percentage of fibrinogen adsorption increases with time, suggesting the formation of a protein corona around the nanoparticle [2,37].

### 3.2. ZnO Albumin Interactions

#### 3.2.1. UV-Visible Spectrophotometry

The interaction between albumin and ZnO was studied by using absorption spectroscopy and fluorescence techniques, the most powerful techniques to investigate the interaction between nanostructured inorganic particles and biological molecules. Bovine serum albumin (BSA) was used to study this potential interaction [38].

UV-Vis spectroscopy was used to investigate the interaction between BSA and the ZnO particles; different assays were performed in the absence and presence of ZnO NPs at different concentrations. As shown in Figure 4, the absorption band of BSA at 280 nm slightly increases with the concentration of ZnO. Since ZnO NPs do not display an absorption band around 280 nm, the increase in the intensity of the peak at 280 nm for the BSA with the addition of ZnO NPs can be attributed to the formation of a complex because of the ZnO-BSA interaction [38].

Another highlight is that the interval between the concentrations increases significantly as the particle size decreases. The absorbance spectra at 30 min show that the band at 280 nm increases with the concentration of 50 nm ZnO more than that obtained with 100 nm ZnO, and this lasts longer than that of micro-ZnO; in that case, absorbance was almost invariable. As shown in the figure, the behavior at 18 h is similar, and it does not provide additional information from that obtained at 30 min. Therefore, it could be said that the absorbance not only depends on the concentration but also on the particle size since the behavior of the three zinc oxides is slightly different. On the other hand, incubation time has no influence on the BSA absorbance. This behavior has been observed in other cases where different concentrations of ZnO have been used [38].

Other studies performed in the literature use the variation of the λmax of ZnO NPs at 375 nm by adding different amounts of BSA. The absorbance variations observed with this procedure also reveal the formation of complex BSA-ZnO particles [39].

#### 3.2.2. Fluorescence Measurements

The fluorescence spectra (Figure 5) obtained after incubating BSA (2 mg/mL) with different concentrations of ZnO (0, 0.8, 1.6, 2.5, and 3.3 mg/mL) show that BSA has a strong emission band at 340 nm when excited at a 278 nm wavelength (blue line). The presence of ZnO particles causes a decrease in fluorescence intensity that is inversely dependent on ZnO concentrations, which suggests the formation of certain complexes created by the binding between ZnO NPs and BSA [38].

Fluorescence quenching refers to any process that decreases the fluorescence intensity of a sample, in this case, albumin. A variety of molecular interactions can result in quenching, including excited state reactions, energy transfer, ground state complex formation, and collisional quenching [20]. Depending on the kind of interaction between the quencher and BSA, fluorescence quenching can occur through two different mechanisms, which are usually classified as dynamic and static quenching. Dynamic quenching occurs when the excited-state fluorophore (a fluorescent chemical compound that can re-emit light upon light excitation) of the BSA is deactivated upon contact with the quencher molecule in solution. Static quenching occurs because of the formation of a nonfluorescent complex between the fluorophore and the quencher [38]. In this case, the interaction between ZnO and BSA occurs through a static quenching mechanism.

In almost all the referred studies, fluorescence quenching is studied at different temperatures. As the temperature increases, the stability of the complexes decreases, and, therefore, the static quenching is attenuated. This is because the binding that occurs between ZnO and BSA at high temperatures to form a complex is destabilized, and then, this unstable complex can decompose [38].

As mentioned above, ZnO interacts with albumin, reducing the fluorescence intensity. The shoulder that appears at about 377 nm becomes slightly clearer with the addition of particles, and this could be attributed to a partial denaturation of the protein or to a protein conformational change. Since this change has not been observed in any of the other metal oxides tested, it appears that ZnO NPs act as a denaturing agent. Other NPs such as TiO_2_, CeO_2_, and Al_2_O_3_ also cause an important decrease in the fluorescence intensity of the BSA, although alumina is the most inert metal oxide NP [40].

Of the two techniques used to study the interaction between albumin and zinc oxide, both reveal the formation of a BSA-ZnO complex, but the variations are much more important in terms of fluorescence, demonstrating that this technique is the best way to study this phenomenon.

### 3.3. Hemolytic Activity of Zn

#### 3.3.1. Media Interaction on Hemoglobin Adsorption

Previous studies with different metallic nanoparticles failed to determine the potential hemolytic activity of ZnO indicating its high binding affinity with hemoglobin [18,41]. In this sense, the ability of ZnO nanoparticles to capture hemoglobin protein was assessed using three different solutions (PBS, 0.9% NaCl, and tris–maleate) at two different pH values (pH 7.4 and pH 5.7). Moreover, for comparative purposes, Al_2_O_3_ was also included. After incubating the nano- and micro-sized metal oxides, the disappearance of the characteristic red color provided by the hemoglobin in solution was observed. In Table 3, a comparison of the behavior of both metal oxides in different media and pH conditions is presented. The data indicate that the adsorption of hemoglobin was only recorded for zinc oxide in tris–maleate, independently of the pH of the buffer solution, and in 0.9% NaCl at pH 5.7. Moreover, this behavior is not dependent on the ZnO particle size. For this reason, posterior hemolytic studies were performed in PBS at pH 7.4.

Based on these interferences, we conclude that the selection of adequate working media to study the hemolytic activity of metal oxide nanoparticles is crucial for a good interpretation of the data and depends on the characteristics of the specific metal. As an example, Figure 6 shows how ZnO adsorbs hemoglobin protein in acidic 0.9% NaCl (pH 5.7), whereas this behavior was not observed for Al_2_O_3_.

This phenomenon could be explained by the importance of pH in hemoglobin protein behavior. The isoelectric point of hemoglobin is 6.8, and the isoelectric point of ZnO is 7.3, but the pH of the 0.9% NaCl medium is 5.7. For this reason, we hypothesize that at this acidic pH, hemoglobin has a positive charge, while, according to the literature, the zeta potential of zinc oxide at this same pH is 26.27 (not demonstrated in 0.9% NaCl). Thus, the adsorption may be due to electrostatic attraction between particles; however, to corroborate this hypothesis, the zeta potential of the zinc oxide particles in 0.9% NaCl should be determined.

#### 3.3.2. Hemolytic Activity in PBS Media, pH 7.4

Effect of temperature

For particles in contact with the human body, it is essential to understand their behavior under physiological conditions. For this reason, we studied the potential hemolytic activity effects of the three zinc oxides (50 nm, 100 nm, and micro) at various concentrations (0.2, 0.4, 0.8, and 1.6 mg/mL) with human RBC at room temperature (RT) and the physiological body temperature, 37 °C (Figure 7).

At RT, it is observed that there is an increase in hemolysis in the presence of micro-ZnO and 100 nm ZnO in parallel with the concentration assessed, with maximal hemolysis recorded at 0.8 mg/mL. In contrast, for 50 nm ZnO particles at the maximal concentration studied, less than 20% hemolysis is recorded. In the case of 37 °C, maximal hemolysis was attained in all three cases at 0.8 mg/mL, followed by a decrease at 1.6 mg/mL. However, previous studies by our group revealed that, from the hemoglobin spectra, this protein suffers denaturalization (Appendix A), and thus, the hemolytic activity of these metal oxide nanoparticles can be underestimated. Studies from [33] comparing the potential toxicity of different metal oxides indicate that 100 nm ZnO was the most toxic of these metal oxides. Here, we found that these nanoparticles are not more toxic than the micro-sized material. Thus, low concentrations of ZnO, which are intended to be administered as a drug, would not be damaging [42].

The results obtained at room and physiological temperatures show that hemolysis increases with increasing temperature [43]. This observation suggests that red blood cells at RT can behave differently than expected because not only is the temperature lower than the physiological one but there are also greater fluctuations during the day. Moreover, these temperature fluctuations correlate well with the highest error bars encountered, especially in the case of 100 nm ZnO particles.

Effect of albumin

The hemolytic activity of ZnO particles when albumin protein is included in the media (Figure 8) shows similar behavior to that of pH 7.4 PBS alone. At RT, hemolysis increases in a dose-dependent manner, and more dispersion is observed, while at physiological temperature, hemolysis decreases at 1.6 mg/mL, indicating that the hemoglobin is suffering denaturation, as stated before. The data values are also like those in Figure 7, and no statistical differences were found when comparing hemolysis without albumin. These results suggest that higher concentrations of albumin are required to interact with ZnO nanoparticles [36] and that a protein corona is not formed at 0.5 mg/mL of BSA.

Thus, it can be concluded that ZnO causes dose-dependent hemolytic activity that also depends on temperature. Although the mechanism of membrane perturbation by ZnO is not clear [44], these data suggest that ZnO can enter cells, induce morphologic alterations, and subsequently produce a rupture in the cell membrane.

#### 3.3.3. Effects of ZnO on Erythrocyte Morphology

The interaction between ZnO particles and human erythrocytes was observed with scanning electron microscopy (SEM). Healthy and young erythrocytes have a biconcave disc-shaped morphology called the discocyte. Alterations in cytoskeletal structures result in membrane deformability, which leads to modified shapes such as echinocytes, erythrocytes with abnormal cell membranes characterized by many evenly spaced thorny projections [45]. Incubating erythrocytes for 3 h in the presence of ZnO induces an increase in the number of echinocytes at different stages. However, the presence of late echinocyte-shaped spherical bodies with spicules distributed over their surfaces is easily identified (Figure 9, white arrows). At 24 h of incubation, the number of red blood cells decreases because ZnO particles (nano and micro) cause ruptures in the membrane, as mentioned before, but crenated-shaped echinocytes are still observed (Figure 9, white arrows). In these images, the effect of time incubation is more evident in the case of nanosized particles than in microparticles.

### 3.4. Coagulation Studies and Plasma Protein Interactions

#### 3.4.1. Coagulation: Prothrombin Time and Activated Partial Thromboplastin Time

Prothrombin time (PT) and activated partial thromboplastin time (aPTT) were used to evaluate the extrinsic and intrinsic pathways of coagulation, respectively [46]. The purpose of this test is to determine how ZnO particles affect the coagulation time if they reach body fluids. Figure 10 shows the effect of the three zinc oxides on the intrinsic and extrinsic pathways of coagulation.

The data obtained for PT at the highest concentration assessed shows that, in the case of the intrinsic route, coagulation time increases when the diameter of the particle is greater (micro-sized particles), unlike the extrinsic route. Previous studies have shown that blood clotting is closely related to particle size. This was the case with silicon NPs, where the intrinsic pathway of coagulation increased significantly as the size of these particles increased. The results obtained showed that the small-sized particles were practically inert, while the larger ones did present significant differences when determining coagulation [47].

As the concentration of the different zinc oxides increases, the coagulation time, both in the extrinsic (PT) and intrinsic (aPTT) pathways, also increases. In the case of aPTT, this increase is already evident at a concentration of 0.5 mg/mL, in particular for 100 nm ZnO and micro-sized particles, although it is not statistically different from the control samples. Statistical analysis also revealed that nanoparticles affect coagulation time in a similar way to ZnO micro-sized particles and are slightly less toxic at the highest concentration assayed.

Some authors suggest that the fact that the intrinsic pathway is more affected than the extrinsic pathway is probably due to the structural loss of factor XII (FXII, a key pro-enzyme of the coagulation cascade) when it comes into contact with the particles [48], that is, a denaturation caused by structural and functional changes of the native protein [47].

#### 3.4.2. Two-Dimensional Gel Electrophoresis of Human Plasma Protein: SDS-PAGE

Differential behavior between particles can be attributed to their size, particle curvature, and surface area [49]. This statement agrees with the amount of protein determined from the particle’s pellet, presented in Figure 11A, as a clear correlation between total protein adsorbed and particle size is observed; as the particle size diminishes, surface area and particle curvature increase, and so does the total amount of protein adsorbed.

Herein, only the most abundant proteins with characteristic and easily recognizable stained bands, such as albumin and fibrinogen, could be reliably identified (Figure 11B). Our findings show that certain proteins were adsorbed on particles with greater affinity than others (see Appendix A). For instance, in lane 72, which corresponds to a 50 nm particle’s pellet, subunits β and γ [50] can be easily identified among the stained protein bands. Moreover, these bands present more intense staining than the one that corresponds to albumin, meaning that fibrinogen is present in a greater concentration in the protein corona. In normal conditions, human plasma albumin is found in concentrations 20 times higher than fibrinogen, as can be observed in lane 1 (human plasma), which reveals a great stain band mainly constituted by albumin, while the ones from fibrinogen β and γ remain regular-sized. In conclusion, it can be assumed that fibrinogen has more affinity for ZnO particles than albumin, especially 50 nm ZnO. It should be noted that the albumin/fibrinogen relationship varies with particle size: lane 5 (micro-sized ZnO) shows more albumin than fibrinogen.

Fibrinogen’s presence in the protein corona could explain the delays in plasma coagulation reported here; if fibrinogen is trapped in the protein corona, it cannot perform its coagulation role, as ZnO particles could be deactivating or denaturalizing it, as previous studies have reported happens to numerous proteins [10]. However, this would imply that the particles that adsorbed the most fibrinogen caused the greatest delays in plasma coagulation, which contradicts our findings for PT and aPTT. Micro-ZnO seems to adsorb less fibrinogen, although its effects on plasma coagulation were more significant than those observed for 50 nm ZnO, which seemed to adsorb more fibrinogen. This suggests the importance of further analysis and gel digestion in order to accurately know each protein that was adsorbed on the particles, and it could alter physiological processes such as plasma coagulation.

Differences between particle sizes were also well observed. For instance, lane 7 from Figure 11B reveals much greater fibrinogen β and γ stains than lanes 5 and 6, meaning that fibrinogen has much more affinity for 50 nm ZnO than it does for the other forms. Moreover, most stains seem to fade as particle size grows, as the global protein–particle affinity lowers with larger particles (Figure 11A), although this is not the case for the albumin/fibrinogen α stain, which seems to gain significance.

#### 3.4.3. Observation of Protein Corona by TEM

TEM observations revealed protein accumulations around our particles, as can be seen in Figure 12B,C,E,F,H,I. Black spots correspond to particles, and shadowy staining corresponds to plasma proteins. These findings should be compared to the images in Figure 12A,D,G.

Organic substances were degraded by the electron beam while the observation took place; as time went by, the nanoparticles seemed to move as the surrounding protein media that coated them altered. Altogether, this degradation process proved the presence of protein. The protein presence was corroborated by an elemental analysis (EDX) using the same TEM instrument with the presence of tungsten in the shadowy staining around the particles (Appendix A).

Differences between particles were significant. The 50 nm ZnO NPs revealed large and dense stains from proteins where our particles were immersed, preventing us from finding them sometimes. A similar pattern was observed in the case of 100 nm NPs, although protein stains were lighter and clearer. On the other hand, micro-sized particles revealed a clear protein coating, and no individual protein stains were found. These phenomena agree with the total protein adsorbed, calculated after incubation with human plasma in Section 3.4.2.

### 3.5. Cytotoxicity of ZnO Particles in HaCaT and A549

The purpose of cytotoxic studies is to predict the toxicity of a certain substance in cells. Working with nanomaterials can pose a risk to the worker; since they are such small particles, they can be suspended in the air and, consequently, can be easily inhaled and, therefore, deposited in the respiratory tract, crossing the lung epithelium or penetrating the skin’s protective membrane, causing an inflammatory response or other possible adverse effects [51]. Based on this explanation, in vitro cytotoxicity was determined in two human cell lines: keratinocytes (HaCaT) and lung epithelial cells (A549).

Cytotoxicity was assessed after exposing the two cell lines to serial concentrations of ZnO particles for 24 h via MTT, NRU, and LDH. The MTT assay allows for the evaluation of the metabolic activity of mitochondria, while the cellular property that determines the uptake of NR and the release of lactate dehydrogenase is the integrity of the lysosomal and plasma membranes, respectively [52]. Figure 13 shows HaCaT cell viability in the presence of the three types of zinc oxides. It can be deduced that these products mainly affect mitochondrial activity (MTT) and the plasma membrane (LDH). The lysosomal membrane is practically unaltered; only a slight decrease can be observed at the highest concentration.

To avoid potential interferences of the NPs with the final spectrometric readouts of LDH, we followed the recommendations of [28]. Moreover, previous studies also indicated the suitability of MTT and NRU, as no interferences were found [2,53].

Cytotoxicity is demonstrated by a decrease in cell viability. Thus, in the MTT assay, cytotoxicity increases as the concentration does in a dose-dependent manner [54]. This fact is more visible if we look at a certain concentration; for example, at 100 μg/mL, micrometric ZnO produces a reduction in mitochondrial function by about 80% compared with 50 nm ZnO, which causes a 60% reduction; on the other hand, at a concentration of 0.78 μg/mL, the effects are practically insignificant with respect to mitochondrial function [55]. With reference to particle size, 50 nm ZnO has the least toxicity, while micrometric ZnO at concentrations greater than 25 μg/mL is the most cytotoxic [2].

This behavior is very similar to the LDH test; the higher the concentration, the more lactate dehydrogenase is released into the medium and, therefore, the lower the cell viability. Regarding the size of the particles, it can be observed that the micrometric size causes greater damage to the plasma membrane than 100 nm ZnO, and this is greater than 50 nm ZnO.

There are studies showing that some ZnO NPs after 6 h of incubation already show a decrease in mitochondrial function and LDH release [55]. Other articles also suggest that the size of the particles is directly related to their cytotoxicity [56].

In Figure 14, it is shown that A549 cell viability was treated with zinc oxide after 24 h of exposure and obtained using the three different methods. In the same way as HaCaT, the lysosomal function of the lung cells only presents significant differences at concentrations higher than 50 μg/mL [52].

In the case of the MTT test, it is also observed that the higher the concentration, the lower the cell viability. Then, if we analyze the percentage of living cells at 100 μg/mL, all the compounds affect mitochondrial activity practically equally (the functional reductions in the mitochondria of 50 nm, 100 nm, and micrometric ZnO are 60%, 52%, and 56%, respectively).

In the same way, the test evaluates the integrity of the membrane has a dose-dependent effect, wherein the higher concentration presents a lower viability since the amount of enzyme released into the medium is greater, and, therefore, there is a greater number of damaged cells. In this case, micrometric zinc oxide has the greatest cytotoxicity, followed by 100 nm ZnO and 50 nm ZnO. There are studies that postulate the idea that the integrity of the plasma membrane depends directly on the structure of the compound studied [52], but here, the most distinctive feature is particle size.

Based on the cytotoxicity data, the inhibitory concentration (IC_50_) was calculated, which allows us to make comparisons between the different products and between the different cell lines in the most effective way. This value is obtained by fitting the data obtained to the best curve. In Table 4, the IC_50_ of the three zinc products in the two cell lines (HaCaT and A549) determined using the MMT and LDH methods can be compared. In the case of the NRU test, the IC_50_ could not be calculated since, at the maximum tested concentration (100 μg/mL), cell viability is approximately 80% in both cell lines. It is observed that the IC_50_ of all zinc oxides is lower than 100 μg/mL. Regarding the MTT and LDH tests, if we compare the different zinc oxides in the keratinocyte cell line, it can be seen, as we mentioned previously, that micrometric ZnO is the most cytotoxic, followed by 100 nm ZnO and 50 nm. In the case of lung epithelial cells, 50 nm ZnO is also the least cytotoxic. If we compare the cell lines, the integrity of the cell membrane in the case of 50 nm ZnO is more affected in the keratinocyte line than in the lung epithelial line.

## 4. Conclusions

In conclusion, the study presented in this research provides valuable insights into the behavior of ZnO nanoparticles (NPs) in physiological media and their impact on various biological processes. Despite challenges in characterizing ZnO NPs in PBS, it was observed that ZnO NPs tend to form aggregates and agglomerates. However, the presence of proteins could potentially avoid NP agglomeration and aggregation effects, as can be concluded from the decrease in the hydrodynamic diameter and protein adsorption capacity of nanoparticles with time, with fibrinogen being the most adsorbed protein. The interaction between albumin and ZnO NPs was confirmed through absorption and fluorescence techniques, with fluorescence results proving to be more accurate.

Moreover, our study also highlighted that pH and media can interfere with hemolysis determination, leading to misinterpretations if potential hemoglobin adsorption is not previously assessed. Based on this, pH 7.4 PBS is an adequate medium to use to conclude that the hemolytic activity of ZnO NPs is dose- or temperature-dependent, with no evidence that nanosized ZnO is more hemolytic than micro-sized particles. However, the presence of 0.5 mg/mL of albumin failed to prevent this hemolytic activity, suggesting that a “protein corona” may not form under these conditions, although previously, the formation of a complex between albumin and ZnO NPs was described at the highest albumin concentration.

The coagulation process was also affected by ZnO NPs, with the extrinsic pathway being the most influenced. As stated, ZnO NPs can agglomerate rapidly in aqueous media, but agglomeration was mitigated in plasma. The presence of a protein corona on ZnO NPs was visually confirmed through TEM observations, and the hydrodynamic diameter of ZnO NPs in the presence of proteins was found to decrease, avoiding agglomeration effects as described when ZnO NPs were in the presence of fibrinogen alone.

Cytotoxicity studies revealed that ZnO NPs exhibited similar behavior during the MTT and LDH methods, with micrometric-sized particles being more cytotoxic and exhibiting a lower IC_50_. Overall, a relationship between nanometer size and toxicity cannot be directly attributed.

In summary, the results of this study highlight the complex interactions between ZnO NPs and biological systems, including their aggregation behavior, hemolytic activity, protein corona formation, coagulation effects, and cytotoxicity. These findings contribute to a better understanding of the potential impacts of ZnO NPs on human health and suggest the need for further investigations to elucidate the underlying mechanisms and optimize the safe use of ZnO NPs in various applications.

## Figures and Tables

**Figure 1 nanomaterials-13-01800-f001:**
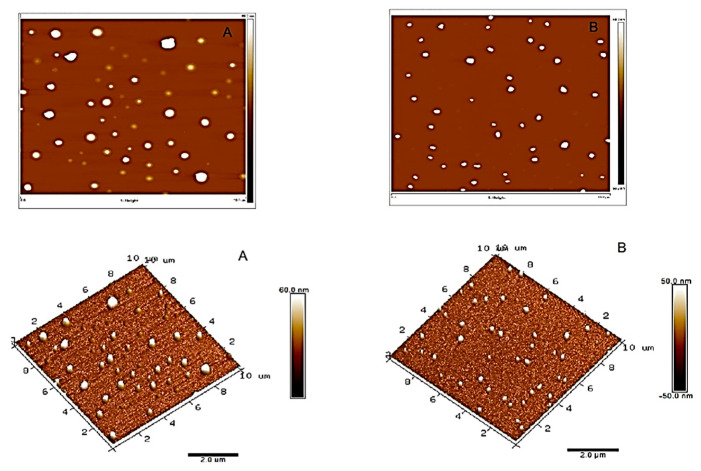
Topographical AFM images (2D and 3D) and particle size histograms of ZnO 50 nm (**A**) and 100 nm ZnO (**B**).

**Figure 2 nanomaterials-13-01800-f002:**
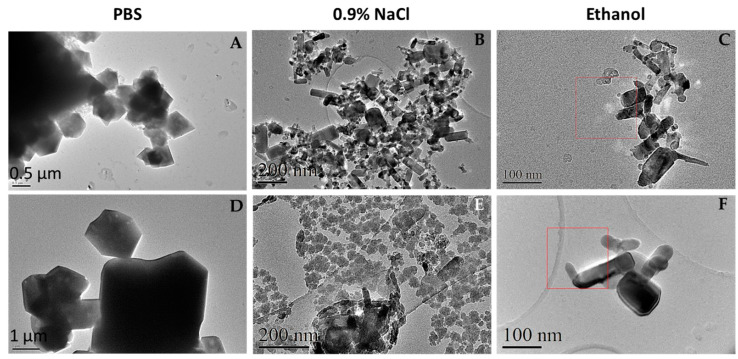
TEM microphotographs of commercial ZnO nanoparticles in different media. ZnO nanoparticles of 50 nm (**A**–**C**) and 100 nm (**D**–**F**) were suspended at a final concentration of 100 μg/mL in PBS (**A**,**D**), 0.9% NaCl (**B**,**E**), and ethanol (**C**,**F**). The suspension was obtained from the dilution and ultrasonication of a previously ultrasonicated suspension of 1 mg/mL of the sample. Observation and analysis were carried out using a JEOL JEM LaB6-2100 microscope with one drop deposited and dried on a Holey Carbon-Cu grid at CCiT UB.

**Figure 3 nanomaterials-13-01800-f003:**
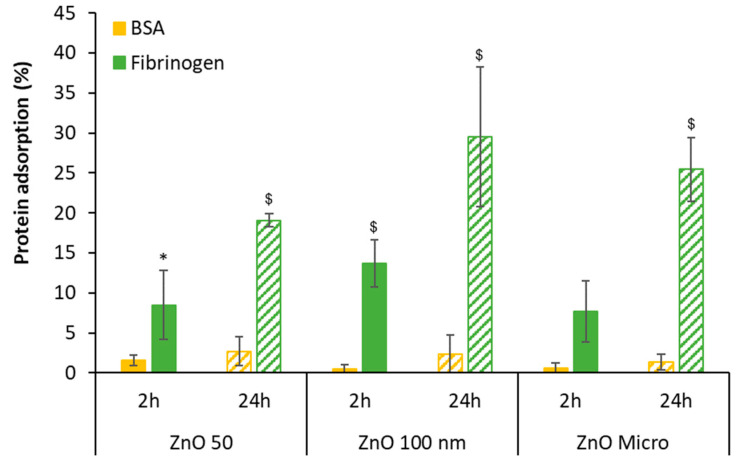
Particle protein adsorption after 2 and 24 h of incubation. Results are expressed as mean ± standard error of *n* = 3 independent experiments. For statistical analysis, an unpaired Student’s *t*-test was used assuming differences for *p* < 0.05; * denotes statistical differences between 2 and 24 h for the same protein; ^$^ indicates statistical differences between media containing BSA versus media containing fibrinogen at the same time of incubation.

**Figure 4 nanomaterials-13-01800-f004:**
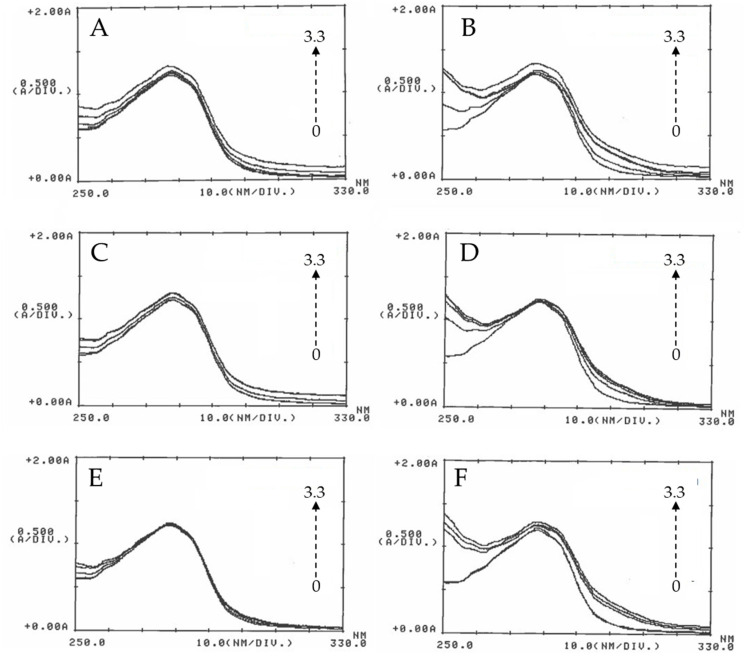
UV-visible absorption spectra of BSA in the presence of ZnO particles. In total, 2 mg of BSA/mL was incubated with 50 nm ZnO (**A**,**B**), 100 nm ZnO (**C**,**D**), and micro-ZnO (**E**,**F**) at concentrations of 0, 0.8, 1.6, 2.5, and 3.3 mg/mL (arrow) at 30 min (**left**) and 18 h (**right**). After incubation, UV-Vis absorption spectra were recorded in a wavelength range of 250–330 nm using a one-centimeter-path-length quartz cuvette.

**Figure 5 nanomaterials-13-01800-f005:**
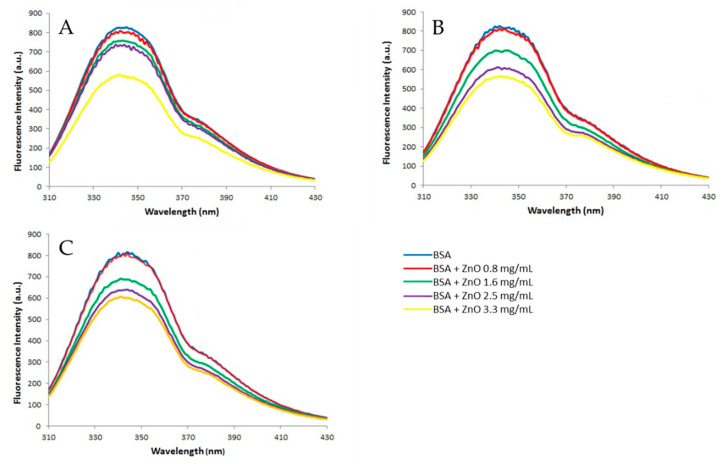
Fluorescence spectra of BSA in the presence of ZnO particles. In total, 2 mg of BSA/mL was incubated with 50 nm ZnO (**A**), 100 nm ZnO (**B**), and micro-ZnO (**C**) at concentrations of 0, 0.8, 1.6, 2.5, and 3.3 mg/mL. Fluorescence measurements were recorded in a wavelength range of 310–430 nm using a one-centimeter-path-length quartz cuvette. The excitation wavelength was set at 278 nm and emission was measured in a range of 275–475 nm.

**Figure 6 nanomaterials-13-01800-f006:**
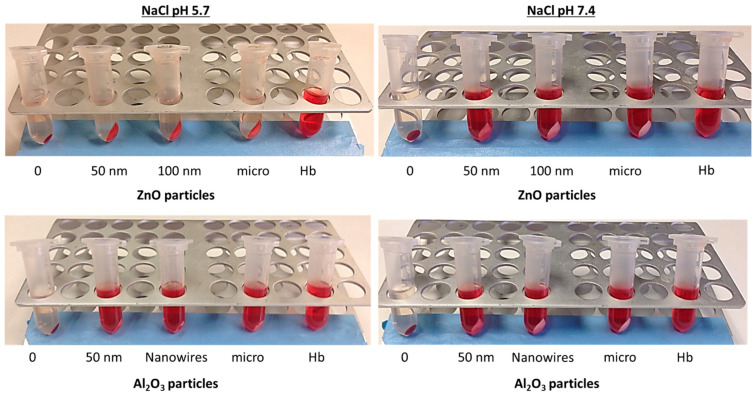
Hemoglobin adsorption in the presence of ZnO and Al_2_O_3_ particles in 0.9% NaCl at pH 5.7 and pH 7.4. A solution of hemoglobin was incubated in the presence of different particles at a concentration of 1 mg/mL for 30 min and then centrifuged at 10,000× *g* rpm for 5 min. Finally, the samples were observed to determine if the hemoglobin was adsorbed by the metal oxide and, thus, if the hemolysis assays could be performed under the conditions studied. RBC incubated with 0.9% NaCl without particles was used as a control for the negative hemoglobin solution (0). Hb: hemoglobin solution incubated with the different media.

**Figure 7 nanomaterials-13-01800-f007:**
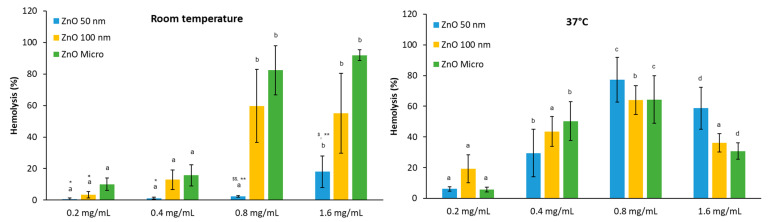
Hemolytic activity of ZnO at different concentrations at room temperature and 37 °C. Results are expressed as mean ± ES for around 4 independent experiments in triplicate. A one-way analysis of variance (ANOVA) and a Scheffé post hoc assay were performed. ^a–d^ denote statistical differences between concentrations at *p* < 0.05 or *p* < 0.01; differences between particle sizes were considered at * *p* < 0.05 and ** *p* < 0.01 between 50 or 100 nm and micro-sized ZnO and ^$^ *p* < 0.05 and ^$$^ *p* < 0.01 between 50 nm and 100 nm ZnO.

**Figure 8 nanomaterials-13-01800-f008:**
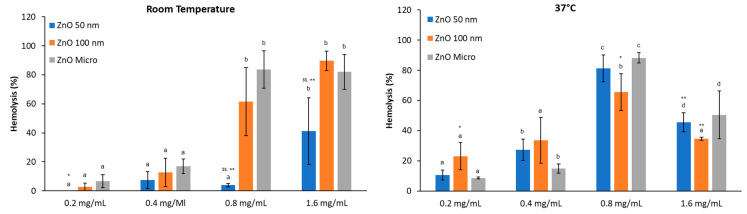
Effect of BSA on the hemolytic activity of ZnO at different concentrations at room temperature and 37 °C. Results are expressed as mean ± ES of around 4 independent experiments in triplicate. A one-way analysis of variance (ANOVA) and a Scheffé post hoc assay were performed. ^a–d^ denote statistical differences between concentrations at *p* < 0.05 or *p* <0.01; differences between particle sizes were considered at * *p* < 0.05 and ** *p* < 0.01 between 50 or 100 nm and micro-sized ZnO and ^$$^ *p* < 0.01 between 50 nm and 100 nm ZnO.

**Figure 9 nanomaterials-13-01800-f009:**
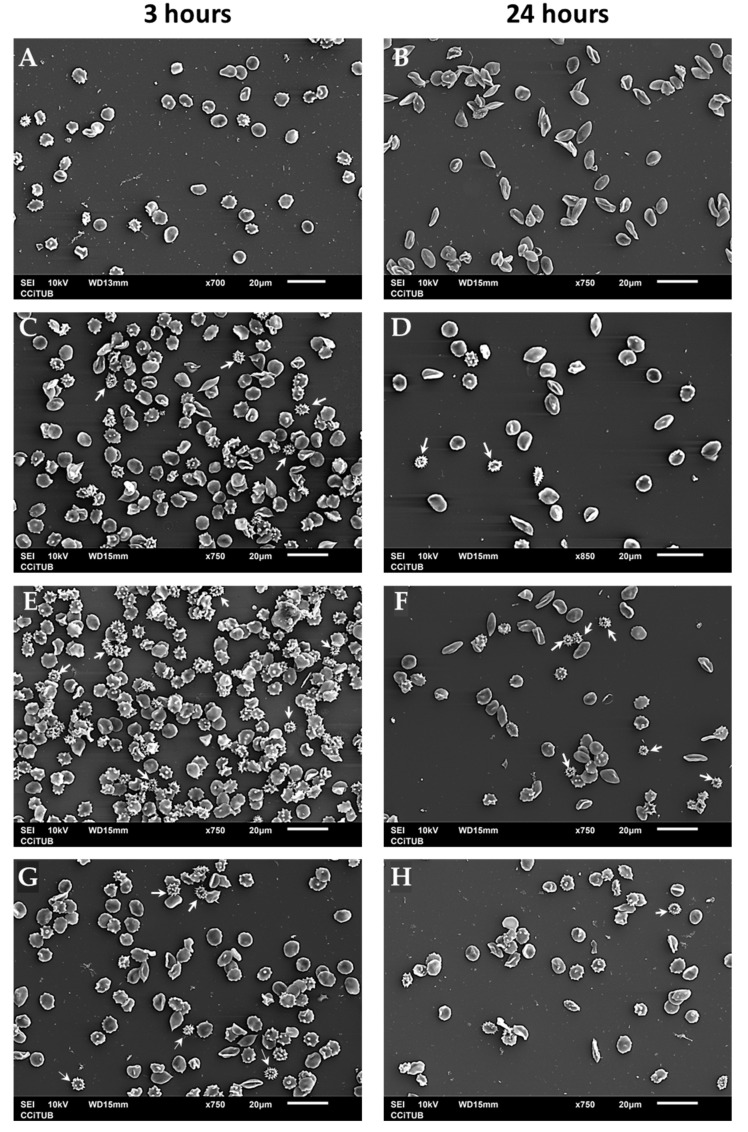
Scanning electron microscopy (SEM) images of human erythrocytes. Erythrocytes were incubated for 3 or 24 h in PBS (pH 7.4) for (**A**,**B**) 0.2 mg/mL 50 nm ZnO (**C**,**D**), 0.2 mg/mL 100 nm ZnO (**E**,**F**), and 0.2 mg/mL micro-ZnO (**G**,**H**).

**Figure 10 nanomaterials-13-01800-f010:**
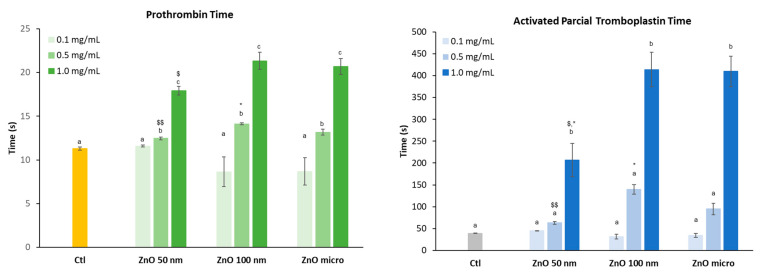
Effect of the different zinc oxides on prothrombin and partially activated thromboplastin time. Results are expressed as mean ± ES of around 3 independent experiments in triplicate. A one-way analysis of variance (ANOVA) and a Scheffé post hoc assay were performed. ^a–c^ denote statistical differences compared with a plasma control at *p* < 0.05 or *p* < 0.01; differences between particle sizes were considered at * *p* < 0.05 between nano- and micro-sized ZnO and ^$^ *p* < 0.05 and ^$$^ *p* < 0.01 between 50 nm and 100 nm ZnO.

**Figure 11 nanomaterials-13-01800-f011:**
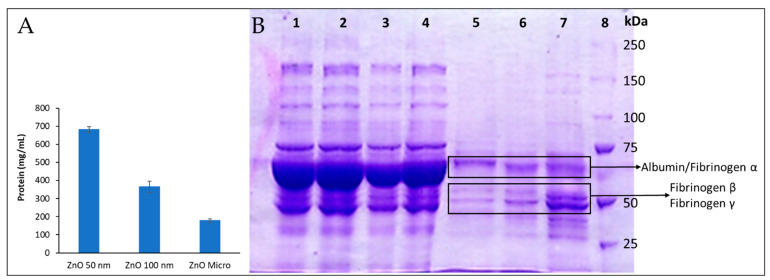
ZnO protein adsorption after incubating ZnO micro- and nanosized particles with human plasma. (**A**): Total protein adsorbed. (**B**): SDS-PAGE electrophoresis. ZnO particles (10 mg/mL) were incubated with human plasma for 30 min and then centrifuged and washed with a solution of sucrose 0.7 M in PBS. The protein extracted (18 µg) was finally run by using a Mini-PROTEAN Tetra Cell unit (Bio-Rad, Hercules, CA, USA). The slab gel consisted of 7.5% polyacrylamide resolving gel and 5% polyacrylamide stacking gel. Electrophoresis was carried out for 10 min at 60 V and then at 200 V. Protein bands were stained with Coomassie Blue R-250 with gentle shaking and destained with a mixture of 7.5% methanol and 10% acetic acid. The molecular weight of the membrane proteins was estimated from the molecular size marker (10–250 kDa). Lane 1: human plasma; lanes 2–4: first supernatant obtained from micro-, 100 nm, and 50 nm ZnO samples after centrifugation, respectively; lane 5: pellet from ZnO microparticles; lane 6: pellet from 100 nm ZnO particles; lane 7: pellet 50 nm ZnO particles; lane 8: molecular weight marker.

**Figure 12 nanomaterials-13-01800-f012:**
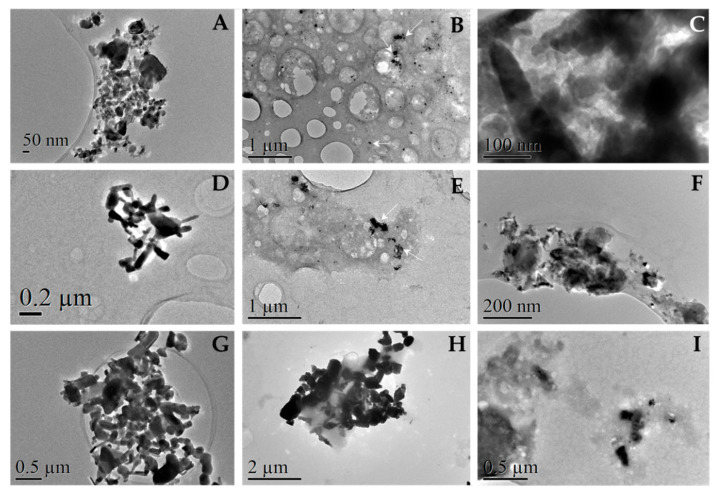
Protein corona images after incubating ZnO particles with human plasma: 50 nm (**A**–**C**), 100 nm (**D**–**F**), or micro- (**G**–**I**) ZnO particles were suspended in MiliQ water before and after 30 min of incubation with human plasma. One sample drop was deposited and dried on a Holey Carbon-Cu grid, and proteins were revealed by adding phosphotungstic acid. Observation and analysis were carried out with a JEOL JEM LaB6-2100 microscope at CCiT UB.

**Figure 13 nanomaterials-13-01800-f013:**
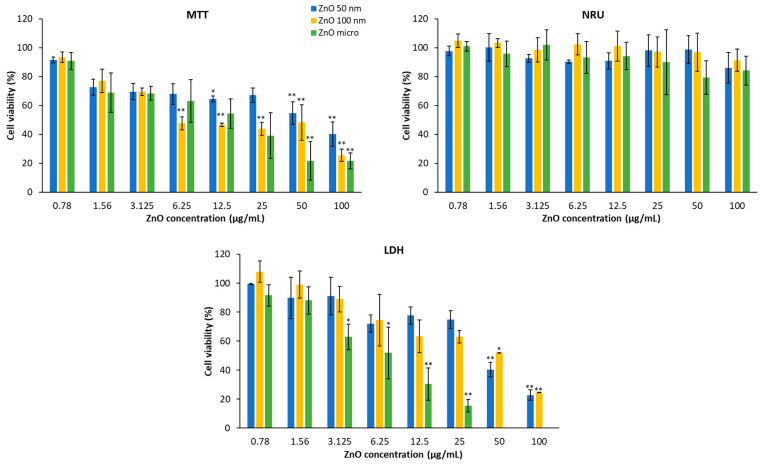
Viability of HaCaT cells in the presence of different concentrations of zinc oxide particles obtained using three different methods (MTT, NRU, and LDH). Results are expressed as the mean ± standard error of *n* = 3 independent experiments. A one-way analysis of variance (ANOVA) and a Scheffé post hoc assay were performed. * *p* < 0.05 and ** *p* < 0.01 denote statistical differences between concentrations.

**Figure 14 nanomaterials-13-01800-f014:**
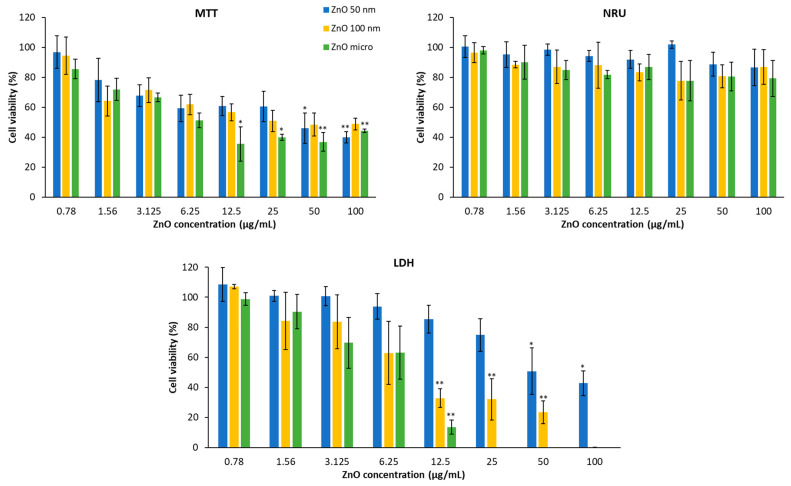
Viability of A549 cells in the presence of different concentrations of zinc oxide particles obtained using three different methods (MTT, NRU, and LDH). Results are expressed as the mean ± standard error of *n* = 3 independent experiments. A one-way analysis of variance (ANOVA) and a Scheffé post hoc assay were performed. * *p* < 0.05 and ** *p* < 0.01 denote statistical differences between concentrations.

**Table 1 nanomaterials-13-01800-t001:** Hydrodynamic diameter (nm) of ZnO NPs in PBS for 2 h and 24 h.

ZnO Particles	2 h ^1^	24 h ^1^
50 nm	1014.9 ± 90.5	2165.0 ± 237.6 ^$$^
100 nm	940.5 ± 72.4	1354.0 ± 62.2 ^$^
Micro	721.1 ± 12.3	916.0 ± 43.2 ^$^

^1^ Results are expressed as mean ± standard error (*n* = 3); for statistical analysis, an unpaired Student’s *t*-test was used assuming differences for ^$^ *p* < 0.05 and ^$$^ *p* < 0.01 with respect to 2 h of incubation.

**Table 2 nanomaterials-13-01800-t002:** Hydrodynamic diameter (nm) of ZnO NPs after incubation in PBS containing 2 mg/mL of bovine serum albumin (BSA) or fibrinogen and DMEM for 2 h and 24 h.

ZnO Particles	2 h ^1^			24 h ^1^		
	BSA	Fibrinogen	DMEM ^2^	BSA	Fibrinogen	DMEM ^2^
50 nm	806.5 ± 52.6 **	30.0 ± 7.2 **	772.2 ± 34.1 **	995.9 ± 171.3 **^,$^	25.6 ± 0.2 **	1199.0 ± 188.2 **^,$$^
100 nm	709.5 ± 15.8 **	53.3 ± 3.6 **	614.90 ± 12.7 **	1334.5 ± 262.3 ^$$^	173.9 ± 50.1 **	827.5 ± 102.0 **^,$^
Micro	625.9 ± 104.7 *	120.4 ± 78.8 **	470.30 ± 20.6 **	1802.0 ± 49.5 **^,$$^	214.0 ± 91.1 **	507.6 ± 36.2 **

^1^ Results are expressed as mean ± standard error of *n* = 3, independent experiments; ^2^ DMEM contains 5% FBS. For statistical analysis, an unpaired Student’s *t*-test was used assuming differences for * *p* < 0.05 and ** *p* < 0.01 with respect to PBS or ^$^ *p* < 0.05 and ^$$^ *p* < 0.01 with respect to the medium itself for 2 h of incubation.

**Table 3 nanomaterials-13-01800-t003:** Hemoglobin adsorption after incubation of metal oxide particles in different media.

Incubation Media	ZnO	Al_2_O_3_
0.9% NaCl pH 5.7 pH 7.4	Adsorption No adsorption	No adsorption No adsorption
PBS pH 5.7 pH 7.4	No adsorption No adsorption	No adsorption No adsorption
Tris–maleate pH 5.7 pH 7.4	Adsorption Adsorption	No adsorption No adsorption

**Table 4 nanomaterials-13-01800-t004:** Inhibitory concentration (IC_50_) for ZnO in HaCaT and A549.

ZnO Particles	HaCaT		A549	
	MTT ^1^	LDH	MTT	LDH
50 nm	68.7 ± 5.9	38.9 ± 1.6	33.1 ± 1.7	91.5 ± 9.3
100 nm	13.1 ± 1.4	16.1 ± 4.7	42.9 ± 5.6	10.3 ± 0.3
Micro	11.3 ± 1.0	6.2 ± 1.3	10.4 ± 0.2	6.3 ± 0.8

^1^ Results are expressed as mean ± standard error of *n* = 3 independent experiments.

## Data Availability

Data are available upon valid request.

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
