# Peer review of "Size Matters? A Comprehensive In Vitro Study of the Impact of Particle Size on the Toxicity of ZnO"

_nanomaterials, 2023, doi:10.3390/nano13111800_

Round 1

Reviewer 1 Report

In their manuscript Mitjans et al. described a comparative in vitro study of the toxicity behavior of zinc oxide (ZnO) nanoparticles and micro-sized particles to understand the impact of particle size on ZnO toxicity, to characterize the particles and their interactions with proteins. Their study provides important insights into the toxicity behavior of ZnO particles. The authors demonstrated the complex interactions between ZnO NPs and biological systems, and that ZnO nanoparticles are not more toxic than micro-sized particles, no direct relationship between nanometer size and toxicity being directly attributed. However, a few observations need to be taken into account, as outlined below:

 1)   The methods and study design are clearly presented, and appropriate for answering the research questions, but some information are lacking, and several  appropriate controls are missing. For example, the viability assays might be influenced by the potential interference of the nanoparticles (ZnO Nanoparticles, micro-sized particles) with MTT or NR. Therefore, the authors should perform an interference test of compounds with the viability dye in the absence of cells. If any compound interferference exists, values shoud be subtracted.

2)   The authors calculated the percentages of cell viability according to the following formula:

Cell viability % = 100 – [(([AC – AB]/[AT-AB])-1) x 100]

(where AC is the absorbance of untreated cells, AB the background absorbance (me-282 dium without cells) and AT the absorbance obtained for treated cells).

The formula for calculating the percentages of cell viability (detected by MTT, NR cytotoxicity assays) is not correct, the numerator and denominator of the fraction are reversed. Therefore all data related to those tests and presented in the Results section might be different. On the other hand, when percentages of cell viability are subtracted from 100%, the resulted values are cell lysis percentages (Cell lysis (%) = 100% - Cell viability (%)).

Please check the formula for calculation of cell viability and the cytotoxicity results.

3) The authors stated that the informed consent and approval were obtained from of healthy donors. The Informed Consent Statement for the healthy donors and Institutional Review Board Statement for ethics should be mentioned also at the end of the manuscript.

4) The manuscript requires proofreading. There are some grammatical errors, and some sentences require re-phrasing.

The manuscript requires proofreading. There are some grammatical errors, and some sentences require re-phrasing.

Author Response

1) The methods and study design are presented, and appropriate for answering the research questions, but some information is lacking, and several appropriate controls are missing. For example, the viability assays might be influenced by the potential interference of the nanoparticles (ZnO Nanoparticles, micro-sized particles) with MTT or NR. Therefore, the authors should perform an interference test of compounds with the viability dye in the absence of cells. If any compound interference exists, values should be subtracted.

Answer: Thank you for the comment. We agree with the reviewer that ZnO particles may present potential interferences with the MTT and NRU assays and therefore lead to misinterpretation of viability absorbances. For these reasons, these interferences have previously been ruled out as indicated in lines 730-732. However. following the reviewer's suggestions, the procedure to rule out these interferences has been added to the material and methods section.

2) The authors calculated the percentages of cell viability according to the following formula:

Cell viability % = 100 – [(([AC – AB]/[AT-AB])-1) x 100]

(where AC is the absorbance of untreated cells, AB is the background absorbance (medium without cells), and AT is the absorbance obtained for treated cells).

The formula for calculating the percentages of cell viability (detected by MTT, NR cytotoxicity assays) is not correct, the numerator and denominator of the fraction are reversed. Therefore, all data related to those tests and presented in the Results section might be different. On the other hand, when percentages of cell viability are subtracted from 100%, the resulting values are cell lysis percentages (Cell lysis (%) = 100% - Cell viability (%)).

Please check the formula for the calculation of cell viability and the cytotoxicity results.

Answer: This formula was only used to calculate cytotoxicity induced by ZnO particles by the LDH assay. In the case of MTT and NRU, cell viability was calculated assuming that untreated cells are 100% viable. This is explained in sentences 294 and 296: Finally, absorbance was measured at 550 nm in a Tecan Sunrise microplate reader (Männedorf, Switzerland), and cell viability was calculated as a percentage considering that untreated cells as 100% viability. However, we have included some clarifications in the material and methods section to avoid misinterpretations.

3) The authors stated that informed consent and approval were obtained from healthy donors. The Informed Consent Statement for the healthy donors and Institutional Review Board Statement for ethics should also be mentioned at the end of the manuscript.

Answer: We have added the Informed Consent Statement and Institutional Review Board Statement at the end of the manuscript.

4) The manuscript requires proofreading. There are some grammatical errors, and some sentences require rephrasing.

Answer: Thanks for the comment. English has been reviewed and modified. Changes are highlighted in red

Reviewer 2 Report

This study describes an in vitro comparative study on the toxic behavior of zinc oxide (ZnO) nanoparticles and micrometer particles. This study aims to understand the impact of particle size on the toxicity of ZnO by characterizing particles in different culture media.

11. The morphology of TEM (Fig.2)is not good enough, please replace it.

22.  Figure 4 is not clear enough, different colors can be used between different lines.

  3.  There are some miscellaneous proteins in Figure 11, can they be purified?

  4.  The research methods for cytotoxicity can refer to the International Journal of Food Microbiology 378 (2022) 109817 and LWT 170 (2022) 114059.

Make minor modifications to the language.

Author Response

This study describes an in vitro comparative study on the toxic behavior of zinc oxide (ZnO) nanoparticles and micrometer particles. This study aims to understand the impact of particle size on the toxicity of ZnO by characterizing particles in different culture media.

1. The morphology of TEM (Fig.2)is not good enough, please replace it.

Answer: Thank you very much for the comment. The nanoparticles of ZnO tend to agglomerate when they are in solution especially if it has many salts. For this reason, the images obtained with TEM and PBS show agglomerates that make it impossible to distinguish the morphology of the nanoparticles properly. In any case, images of ZnO nanoparticles in 0.9% NaCl and ethanol have been replaced to better show nanoparticle morphology.

2. Figure 4 is not clear enough; different colors can be used between different lines.

Answer: Thank you for the comment. Figure 4 are the direct scans obtained from Shimadzu UV-Vis 160, without edition. No colors are presented by the apparatus and the edition of spectra can conduct alterations of the original results. For this reason, the figure was not replaced or changed.

  1. There are some miscellaneous proteins in Figure 11, can they be purified?

Answer: Thanks for the comments, but this is a tentative preliminary study of the effect of the ZnO on blood protein adsorption, then only the more significative representative of blood proteins has been selected.

  1. The research methods for cytotoxicity can refer to the International Journal of Food Microbiology 378 (2022) 109817 and LWT 170 (2022) 114059.

Answer: Thanks for the suggestion. Among the abundant literature dealing with cytotoxicity methods Guadagnini et al (Nanotoxicol. 2015, 9, 13-24) and Zanette, et al (Toxicol Vitro. 2011, 25, 1053–1060) represent well the challenges related to nanotoxicity studies. Therefore, after checking the proposed articles proposed by the reviewer, we consider that these protocols differ from the ones used in our manuscript.

English has been reviewed and modified. Changes are highlighted in red in the manuscript

Round 2

Reviewer 1 Report

The authors answered to all the questions and comments, and added some clarifications to the text of the article. They have also reviewed the English language throughout the manuscript. Therefore, I recommend the publication of the revised article.

 The authors reviewed the English language throughout the manuscript.

Reviewer 2 Report

no problem

no problem